# Rutin Promotes Proliferation and Orchestrates Epithelial–Mesenchymal Transition and Angiogenesis in MCF-7 and MDA-MB-231 Breast Cancer Cells

**DOI:** 10.3390/nu15132884

**Published:** 2023-06-26

**Authors:** Homa Hajimehdipoor, Zahra Tahmasvand, Fatemeh Ghorban Nejad, Marc Maresca, Sadegh Rajabi

**Affiliations:** 1Traditional Medicine and Materia Medica Research Center, Department of Traditional Pharmacy, School of Traditional Medicine, Shahid Beheshti University of Medical Sciences, Tehran 1516745811, Iran; hajimehd@sbmu.ac.ir; 2Traditional Medicine and Materia Medica Research Center, Shahid Beheshti University of Medical Sciences, Tehran 1434875451, Iran; korand2051@gmail.com (Z.T.); parisaghorban015@gmail.com (F.G.N.); 3Aix Marseille Univ, CNRS, Centrale Marseille, iSm2, 13013 Marseille, France

**Keywords:** breast cancer, rutin, epithelial–mesenchymal transition, metastasis, angiogenesis

## Abstract

Rutin has been reported as a potential anti-cancer agent for several decades. This study evaluated the effects of rutin on the proliferation, metastasis, and angiogenesis of MDA-MB-231 and MCF-7 breast cancer cell lines. Increasing concentrations of rutin significantly stimulated the proliferation of MDA-MB-231 and MCF-7 cells compared to controls. Wound scratch assay demonstrated that rutin had an inducing effect on the migration of the cells. In MDA-MB-231 and MCF-7 cells, rutin upregulated MKI67, VIM, CDH2, FN1, and VEGFA and downregulated CDH1 and THBS1 genes. It also increased N-cadherin and VEGFA and decreased E-cadherin and thrombospondin 1 protein expression. Our data indicated that rutin could stimulate proliferation, migration, and pro-angiogenic activity in two different breast cancer cell lines. This phytoestrogen induced invasion and migration of both cell lines by a mechanism involving the EMT process. This suggests that rutin may act as a breast-cancer-promoting phytoestrogen.

## 1. Introduction

Breast cancer is the most common cause of female mortality from cancer around the world [1]. Based on the epidemiological reports during the past two decades, the rate of breast-cancer-related deaths is markedly increasing worldwide [2]. Breast cancer is generally categorized into three different subtypes that include estrogen/progesterone receptor-positive, HER2-positive, and triple-negative forms [3]. Each subtype of breast tumor harbors a specific risk profile. Hence, the patients are treated with a variety of approaches depending on the subtype of tumor, disease stage, and patient preferences [4].

Classical methods of breast cancer therapy such as surgery, radiotherapy, hormone therapy, and chemotherapy are available in clinical settings [4,5]. However, therapy resistance and the development of adverse side effects following the use of conventional therapeutic methods are major drawbacks in breast cancer therapy, causing disease recurrence and relapse. Consequently, there is an urgent need to seek more efficient drugs to treat this disease [6].

Metastasis is a multi-step process during tumor development leading to the spread of cancer cells to the secondary organs or tissues [7]. Breast cancer is a highly metastatic cancer and represents a poor prognosis due to the development of a secondary tumor in the distant organ(s) [8]. When breast cancer cells start to metastasize, they undergo epithelial–mesenchymal transition (EMT) to acquire invasive characteristics [9]. EMT is a process of morphology alteration, by which epithelial cells acquire fibroblast-like characteristics typical of mesenchymal cells via orchestrating a set of transcriptional events [10]. EMT is a pivotal process in morphogenetic alterations during embryonic development, wound repair, and cancer metastasis [11]. Indeed, metastatic cancer cells usurp this developmental process to initiate a multi-step program leading to tumorigenesis and metastasis [12]. Angiogenesis is an essential step in tumor development because tumor growth and metastasis rely on efficient angiogenesis [13].

Rutin is a flavonoid compound found in a variety of plant sources. This compound has various medicinal effects, including anti-inflammatory, antioxidant, anti-diabetic, vascular protective, antimicrobial, and anti-cancer activities [14]. The anti-proliferative and anti-metastatic effects of rutin have been proven in various cancer cells. Indeed, this natural product has been shown to hamper the proliferation and migration of human lung and colon cancer cell lines [15].

In breast cancer cell lines, data on rutin are more confusing. Indeed, one study performed on MB-MDA-231 and MCF-7 cells has shown that rutin does not possess anti-proliferative effect on these cells, at least up to 20 µM, but was able to increase the cytotoxicity of two chemotherapy agents, cyclophosphamide and methotrexate [16]. On another hand, Elsayed et al. reported that rutin inhibited triple-negative breast cancer (TNBC) cell proliferation, migration, and invasion by a mechanism involving c-Met kinase but used hepatocyte growth factor (HGF) to stimulate proliferation and migration of the cells [17]. To the best of our knowledge, no previous study evaluated the impact of rutin on angiogenesis and EMT, as two pivotal steps in the metastasis cascade, in breast cancer cells. Therefore, the current study aimed to evaluate the effect of rutin on these two key processes in the metastasis of two different breast cancer cell lines, MB-MDA-231 and MCF-7.

## 2. Materials and Methods

### 2.1. Cell Culture and Treatments

The breast cancer cell lines used in this present study, MCF-7 and MDA-MB-231, were purchased from Pasture Institute, Tehran, Iran. All the materials used for cell culture were purchased from Gibco, UK. MCF-7 and MDA-MB-231 cell lines were cultured and incubated in a humidified incubator at 37 °C and 5% CO_2_. Rutin was purchased from Sigma-Aldrich (St. Louis, MO, USA) and the stock solution was prepared in dimethyl sulfoxide (DMSO).

### 2.2. Cell Viability

Cells were seeded into 96-well cell culture plates at the density of 10^4^ cells per well and allowed to adhere to the bottom of the wells. After 24 h, the seeded cells were treated with increasing concentrations of rutin (0, 6.25, 12.50, 25, 50, 100, 200, and 400 μM) for 24, 48, and 72 h. The control cell lines were treated with normal culture medium and vehicle (DMSO). Then, the viability of each cell line was assayed using the 3-(4,5-Dimethylthiazol-2-yl)-2,5-diphenyltetrazolium bromide (MTT) method. Briefly, at the end of the treatment period, the cell lines were incubated with 5 mg/mL MTT solution for 4 h. Subsequently, the MTT solution was discarded and the formazan product formed was solubilized with 100 µL of DMSO. After 15 min, the absorbance was read at 570 and 630 nm using a plate reader spectrophotometer (PerkinElmer, Waltham, MA, USA).

### 2.3. Wound Healing Assay

The effect of rutin treatment on the invasiveness and migratory capacity of MCF-7 and MDA-MB-231 cell lines was evaluated by in vitro scratch assay as previously described [18]. Briefly, 2 × 10^5^ cells were seeded in a 6-well cell culture plate and allowed to adhere and grow to 50–60%. Subsequently, two horizontal scratches were created in the center of each well using 200 µL sterile pipette tips. One vertical line was created in each well to take photographs of scratches at the same point. Then, each well was washed with phosphate-buffered saline (PBS) buffer solution (pH~7.4) to remove cell debris. Afterward, three test wells were treated with 200 and 400 μM rutin compound and three control wells were treated with a normal culture medium. Finally, the photographs of the wound-healing capacity of both cell lines were captured at 0 h, 9 h, and 24 h.

### 2.4. Real-Time PCR Analysis

The effect of rutin on the gene markers of proliferation, EMT, and angiogenesis was evaluated by using the Quantitative Real-time PCR technique. Following the treatment of MDA-MB-231 and MCF-7 cell lines with rutin at the concentration of 200 µM for 48 h, the total RNAs were extracted from treated and control cells by an RNeasy mini kit (Qiagen Co., Seoul, Republic of Korea). To assess the quality of total RNA, the absorbance of the samples was measured by a Nanodrop 2000c spectrophotometer. Afterward, 2 μg of total RNA sample was collected to synthesize cDNA molecules using a cDNA Synthesis Kit (BioFact, Daejeon, Republic of Korea). Then, ABI PRISM7900HT (Applied Biosystems, Carlsbad, CA, USA) qRT-PCR detection system with SYBR GREEN PCR master mix (Ampliqon, Copenhagen, Denmark) was used to assess the expression levels of MKI67, CDH1, CDH2, VIM, FN1, VEGFA, and THBS1 genes. Glyceraldehyde-3-phosphate dehydrogenase (GAPDH) gene expression was considered the internal control gene. Relative expression (fold changes) of the above-mentioned genes was calculated using the 2^−ΔΔCT^ formula. Table 1 shows the primer sequences used for the amplification of the the genes.

### 2.5. Western Blot Analysis

Cells were seeded in 6-well culture plates at a density of 2 × 10^5^ cells/well and allowed to adhere overnight. Then, the test cell lines were treated with 200 µM rutin for 48 h. The control cell lines were treated with culture medium. After 48 h, the treated cells were lysed using radioimmunoprecipitation assay (RIPA) buffer (Thomas Scientific Inc., Swedesboro, NJ, USA). The protein content of the whole-cell lysates was measured using the Bradford method [19]. Using SDS-PAGE, 40 µg of total protein was separated, and protein blots were transmitted onto PVDF membranes (Roche, Mannheim, Germany). The membranes were then blocked with 5% BSA (Sigma Aldrich, St. Louis, MO, USA) in 0.1% Tween 20 for 1 h. The blots were incubated with primary antibodies (Santa Cruz Biotechnology, Santa Cruz, CA, USA) against E-cadherin, N-cadherin, Fibronectin 1, Vimentin, ki67, VEGF, Thrombospondin1, and β-actin at 4 °C overnight. After extensive washes, the membranes were exposed to secondary antibodies conjugated with HRP for 1 h at room temperature. After further washes, the blots were visualized by adding enhanced chemiluminescence (ECL) substrate.

### 2.6. Data Analysis

The results of the present study were obtained from three independent experiments. The statistical analyses were performed with GraphPad Prism software version 7. The data were analyzed using the one-way analysis of variance (ANOVA) test followed by Duncan’s multiple range tests as the post hoc. *p*-values less than 0.05 and 0.01 were considered significant. All data are representative of the mean ± standard deviation (SD). Analysis of wound areas was performed by ImageJ software (U.S. National Institutes of Health).

## 3. Results

### 3.1. Effect of Rutin on the Proliferation of MDA-MB-231 and MCF-7 Cell Lines

Treatment of MDA-MB-231 and MCF-7 cell lines with increasing concentrations of rutin for 24, 48, and 72 h promoted the proliferation of both cancer cell lines in a dose-dependent manner in comparison to the culture medium (control) and DMSO-treated cells. As depicted in Figure 1A–C, treating the MDA-MB-231 cell line with rutin for 24, 48, and 72 h significantly increased the number of viable cells at concentrations more than 200 µM.

Figure 2A–C also demonstrate that treatment of MCF-7 cells with increasing concentrations of rutin for 24, 48, and 72 h also significantly enhanced the proliferation of these cancer cells at the dose of more than 200 µM.

### 3.2. Effects of Rutin on the Invasion and Migration of MDA-MB-231 and MCF-7 Cell Lines

To determine the effect of rutin on the invasion and migration of MDA-MB-231 and MCF-7 cell lines, the cells were seeded into 6-well plates and treated with 200 and 400 μM rutin. The results showed that rutin treatment promoted the invasion and migration of MDA-MB-231 cells at both concentrations of this natural compound (Figure 3A). Indeed, in MDA-MB-231 cells, rutin (200 and 400 μM) significantly and time-dependently decreased the empty areas compared with untreated control (Figure 3B).

Similarly, in MCF-7 cells, rutin had a promoting effect on the invasion and migration of the treated cells at concentrations of 200 and 400 μM in comparison to untreated controls (Figure 4A,B). However, the statistical analysis showed that only rutin 400 μM significantly increased the invasion and migration of MCF-7 cells.

### 3.3. Effects of Rutin on Gene and Protein Expression in MDA-MB-231 and MCF-7 Cell Lines

To evaluate the effect of rutin on markers of proliferation, EMT, and angiogenesis, the expression levels of the corresponding gene markers in the MDA-MB-231 and MCF-7 cell lines were quantified. Figure 5A illustrates that rutin (200 μM) significantly amplified the mRNA expression of proliferation marker MKI67, pro-angiogenic marker VEGFA, and EMT markers VIM, CDH2, and FN1 in the MDA-MB-231 cell line in comparison to the untreated control cells. In parallel, rutin downregulated mRNA expression of EMT marker CDH1 and anti-angiogenic marker THBS1 in this cancer cell line. As depicted in Figure 6A, rutin (200 μM) also significantly upregulated the expression of MKI67, as a proliferation marker, in MCF-7. Rutin also activated the EMT process in MCF-7 cells by suppressing the mRNA expression of CDH1 and increasing VIM, FN1, and CDH2 expression compared to the untreated control cells. Rutin at 200 µM also promoted the expression of the pro-angiogenic marker VEGFA in MCF-7 cells with no remarkable effect on the expression levels of the anti-angiogenic gene marker THBS1.

Treatment of the MDA-MB-231 and MCF-7 cell lines with rutin had a modulatory effect on the expression levels of various proteins involved in the angiogenesis and EMT process in these cancer cells. Thus, according to the results of Western blotting analyses, rutin, at the concentration of 200 μM, had a possible effect on the EMT process in the MDA-MB-231 cell line by downregulating E-cadherin and upregulating N-cadherin in comparison to untreated control cells. However, it had no remarkable effect on the expression levels of Vimentin and Fibronectin 1 proteins compared with the controls. This natural compound also showed no significant effect on the protein levels of proliferation marker ki67 in the MDA-MB-231 cell line. Nonetheless, rutin activated angiogenesis by elevating the expression of the pro-angiogenic marker VEGFA and declining the anti-angiogenic marker Thrombospondin 1 (Figure 5B).

Treatment of MCF-7 cells with rutin 200 µM decreased the protein expression of E-cadherin with increased levels of ki67, N-cadherin, and Fibronectin 1 protein compared with the untreated control with no significant effect on the protein level of Vimentin (Figure 6B). As observed with MDA-MB-231 cells, in MCF-7 cells, rutin acted positively on the angiogenesis process by upregulating VEGFA with no remarkable effect on anti-angiogenic Thrombospondin 1.

## 4. Discussion

According to the literature, rutin has shown anti-proliferative and anti-metastatic effects in some cancer cell lines [15]. For breast cancer cells, results were less evident, with one study showing the absence of an effect of rutin alone (at least for a concentration of rutin up to 20 µM) and another study demonstrating an anti-proliferative effect on breast cancer cells pre-treated with hepatocyte growth factor (HGF) [16,17]. Therefore, we aimed to evaluate the potential effect of the rutin flavonoid on the proliferation, invasion, migration, and angiogenesis in two metabolically different human breast cancer cell lines, MDA-MB-231 and MCF-7.

In the present study, the MTT assay results revealed that rutin stimulated the proliferation of both MDA-MB-231 and MCF-7 cancer cell lines at concentrations greater than 200 µM. Evaluation of the expression of ki67 also uncovered that rutin increased this marker in the MDA-MB-231 cell line at the mRNA level but had no effect on the expression of its protein product. Rutin upregulated mRNA and protein levels of ki67 in the MCF-7 cell line. Ki67 is known as a key marker of tumor proliferation and invasiveness [20]. These data may suggest rutin acts as a proliferation stimulator in breast cancer cells.

The results of the wound scratch assay illustrated that rutin acts as an inducer of invasion and migration in the MDA-MB-231 cell line at doses of 200 and 400 µM. However, significant wound closure was observed in MCF-7 cells only after treating the cells with rutin 400 µM. This difference may be due to the higher migratory activity of the MDA-MB-231 cell line in comparison to MCF-7 cells. The wound-closure capacity of rutin may propose a potential metastasis-inducing activity of rutin in breast cancer cells.

Based on the results of the wound scratch assay, we decided to uncover the mechanism by which rutin has promoted the metastasis of MDA-MB-231 and MCF-7 breast cancer cells. Thus, the effect of rutin treatment on the expression levels of EMT markers in both cancer cell lines was evaluated. EMT is the leading event in the process of metastasis that allows an epithelial cell to undergo some biochemical changes leading to a mesenchymal cell phenotype with higher metastatic capacity and invasiveness [21]. EMT is triggered by the downregulation of E-cadherin followed by the upregulation of N-cadherin, Vimentin, and Fibronectin [22]. The data of real-time PCR and Western blotting methods in the present study showed that rutin triggered the EMT process in MDA-MB-231 and MCF-7 cells to promote the invasion and migration of these cancer cells. In MDA-MB-231 cells, rutin decreased the expression of E-cadherin and increased the levels of N-cadherin in comparison to control cells. In the MCF-7 cell line, rutin upregulated N-cadherin and Fibronectin 1 and downregulated E-cadherin. Rutin exerted no effect on the expression of Vimentin in MCF-7 cells and Vimentin and Fibronectin 1 in MDA-MB-231 cells. Rutin had a regulating effect on the expression of the pro-angiogenic marker VEGFA and anti-angiogenic marker Thrombospondin 1 in MDA-MB-231 and MCF-7 cell lines. It increased the expression of VEGFA and decreased the expression of Thrombospondin 1 in the MDA-MB-231 cell line. VEGFA expression was also increased due to the treatment of MCF-7 cells with rutin. However, it had no remarkable effect on the expression levels of Thrombospondin 1. Tumor angiogenesis is a pivotal step in the metastatic propagation of tumor cells to distant organs [23]. Therefore, it appears that rutin could effectively induce the angiogenic pathway to help the progression of breast tumors to secondary organs.

The present data are opposite to those reported in the literature. One possible explanation for these controversial results is based on the fact that rutin is a phytoestrogen compound with a variety of beneficial or adverse effects depending on the subject studied, the sex, the age, and the physiological status [24]. Phytoestrogens have been shown to exert estrogenic effects by acting on estrogen receptors (ERs) [25]. Some studies have proven that rutin elevates the estrogen concentration in plasma and mammary glands and upregulates the expression of ER in some tissues [26]. Synthesis and metabolism of estrogens in steroid-hormone-dependent organs by phytoestrogens may also disrupt the balance of local hormone levels leading to a variety of women’s health issues, varying from altered menstrual cycle to hormone-dependent cancers [25]. Phytoestrogens have been proven to inhibit estrogen inactivation and excretion leading to an increased bioavailability of endogenous estrogens and a disrupted endocrine balance [27]. This pathological alteration may play a critical role in human susceptibility to breast cancer [28]. In a previous in vitro study, the proliferation-stimulating activity of a phytoestrogen compound was evaluated. Yuan et al. provided evidence to highlight that genistein, a phytoestrogen, can induce cell growth in two human breast cancer cell lines (MCF-7 and T47D) in a concentration-dependent manner through ER activation [29]. Genistein promotes the proliferation of these cell lines at lower concentrations but suppresses their growth at higher doses [29]. On the contrary, the present study revealed that rutin phytoestrogen stimulated cell growth at higher concentrations, suggesting that the observed effect is not related to ER activation. Additionally in favor of this hypothesis, rutin was active both on ER/PR-positive MCF-7 and on triple-negative MDA-MB-231 cells, further suggesting that the observed stimulatory effect of rutin does not depend on the steroid receptors.

An alternative scenario may arise from the intracellular metabolism of rutin in breast cancer cell lines. Harris et al. demonstrated that a variety of phytoestrogens act as potent inhibitors of sulfotransferases, which are enzymes that add a sulfate group to estrogens to inactivate them [27]. Moreover, Yuan et al. observed that genistein, a phytoestrogen, was extensively conjugated with sulfate and glucuronide groups in breast cancer cells. Interestingly, these conjugated forms of genistein did not affect the proliferation stimulation of the breast cancer cell lines, and only unconjugated forms played an inducer of proliferation role [29]. Comparing those data with the results of the present study may suggest the inhibition of sulfotransferases by rutin, as phytoestrogen, in MDA-MB-231 and MCF-7 cells, causing the blockage of rutin inactivation. This may have resulted in a high level of the active form of rutin in the intracellular space and the promotion of MDA-MB-231 and MCF-7 cell growth and metastasis. Elsayed et al. conducted a study to evaluate the effect of rutin on the proliferation, migration, and invasion of four different breast cancer cell lines. Opposite to the present study, they reported that rutin could exert anti-proliferative and anti-metastatic effects on breast cancer cells. The comparison between our study and that of Elsayed et al. showed that different results may arise from the differences between the designs of the two studies [17]. Indeed, in their study, Elsayed et al. pretreated the breast cancer cells with HGF as a scatter factor. This growth factor may interfere with the effects of rutin, and pretreatment of the cells with it could render anti-proliferative and anti-metastatic activity to rutin. The use of rutin alone in the present study may be the reason for the opposite results compared to those obtained in the study of Elsayed et al., who used rutin in the presence of HGF.

## 5. Conclusions

The present data unraveled the cancer-promoting effects of rutin on two different breast cancer cell lines. Contrary to the previously published data showing the anti-proliferative effect of rutin on HGF-treated breast cancer cells, the present study demonstrates that this natural phytoestrogen stimulated the proliferation, invasiveness, and pro-angiogenic capacity of breast cancer cells. Rutin induced invasion and migration of these cancer cell lines by a mechanism that involved the EMT process. It also stimulated the angiogenic pathway involved in the metastasis of breast cancer cells. Although rutin has been reported as an anti-cancer agent in the literature, these data suggest that it could act as a breast-cancer-promoting or progressing natural compound. Although the present data is primary, it may pave the way to design more comprehensive studies for the evaluation of the effects of rutin phytoestrogen in a variety of cancers.

## Figures and Tables

**Figure 1 nutrients-15-02884-f001:**
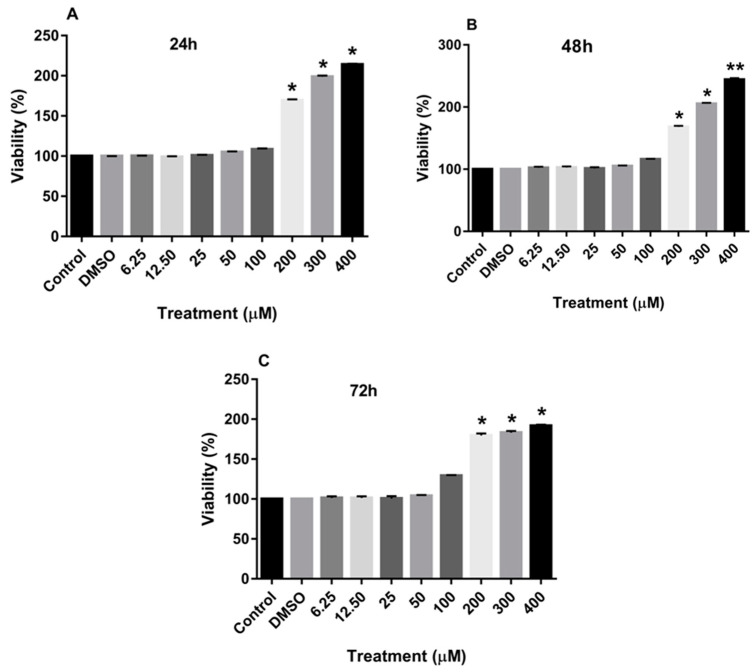
The effects of rutin on the proliferation of MDA-MB-231 cell line. The cells were treated with increasing concentrations of rutin (0, 6.25, 12.50, 25, 50, 100, 200, and 400 μM) for 24 (**A**), 48 (**B**), and 72 (**C**) h. The viability of the cells was estimated by MTT assay. The data are expressed as mean ± SD of three independent experiments. Significant differences were found between the treatment and control groups at * *p* < 0.05 and ** *p* < 0.01.

**Figure 2 nutrients-15-02884-f002:**
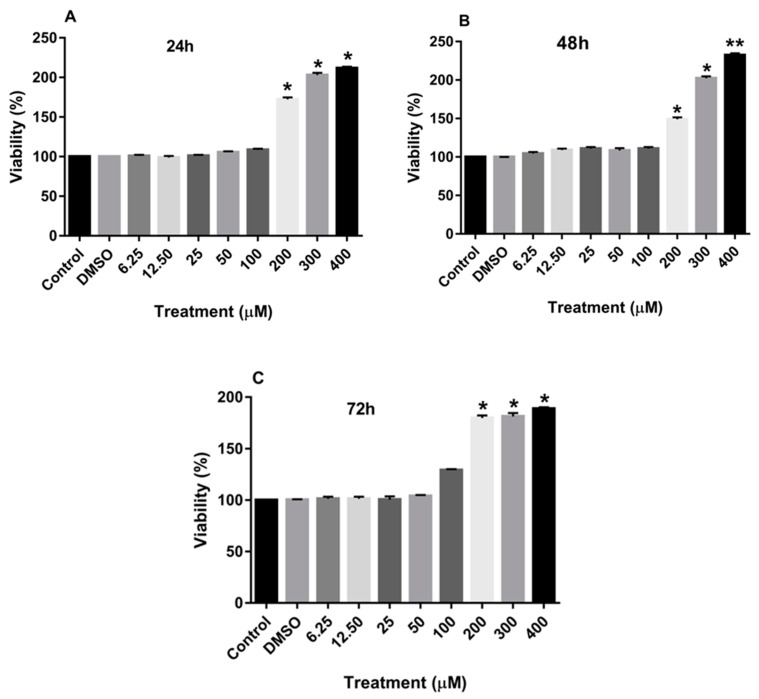
The effects of rutin on the proliferation of the MCF-7 cell line. The cells were treated with various concentrations of rutin (0, 6.25, 12.50, 25, 50, 100, 200, and 400 μM) for 24 (**A**), 48 (**B**), and 72 (**C**) hours. The cell viabilities were determined using the MTT assay method. The data are expressed as mean ± SD of three independent experiments. Significant differences were found between the treatment and control groups at * *p* < 0.05 and ** *p* < 0.01.

**Figure 3 nutrients-15-02884-f003:**
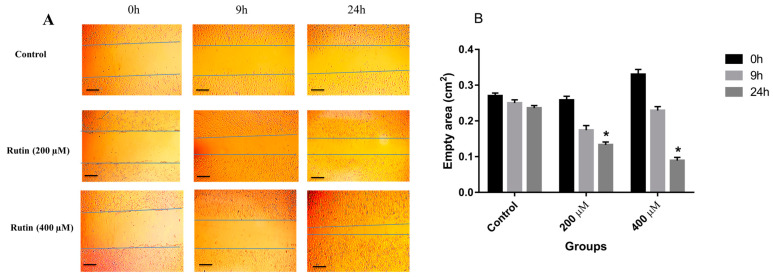
(**A**) The effect of rutin treatment on the migratory capacity of the MDA-MB-231 cell line. The cultured cells were scratched with a sterile pipette tip followed by the treatment with 200 and 400 µM rutin. Photographs of wound-closure process were captured at 0 h, 9 h, and 24 h post-treatment. (**B**) Data are expressed as cell migration area (cm^2^) compared with control (mean ± SD). * *p* < 0.05 vs. control. The scale bar represents 100 µm.

**Figure 4 nutrients-15-02884-f004:**
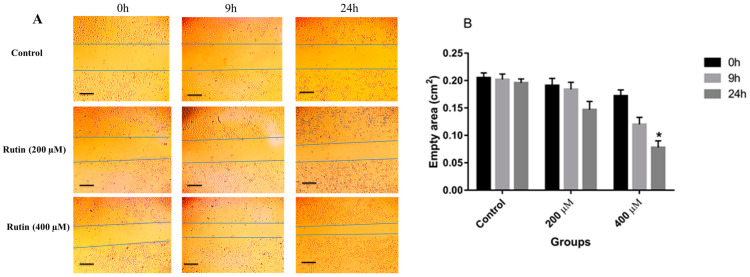
(**A**) The effect of rutin treatment on the migratory capacity of the MCF-7 cell line. The cultured cells were scratched with a sterile pipette tip followed by the treatment with 200 and 400 µM rutin. Photographs of wound-closure process were captured at 0 h, 9 h, and 24 h post-treatment. (**B**) Data are expressed as cell migration area (cm^2^) compared with control (mean ± SD). * *p* < 0.05 vs. control. The scale bar represents 100 µm.

**Figure 5 nutrients-15-02884-f005:**
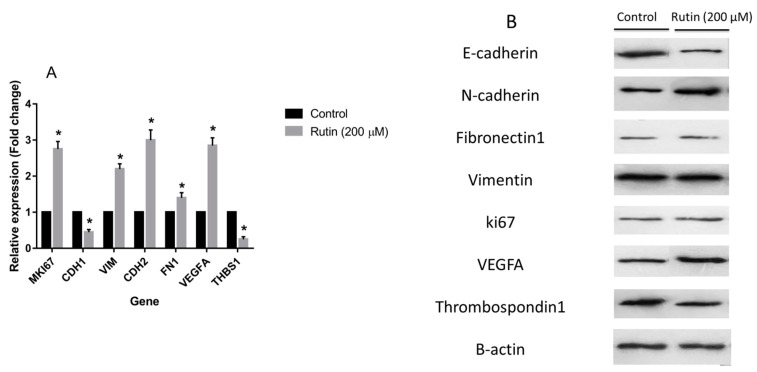
(**A**) The effect of rutin at the concentration of 200 µM for 48 h on the mRNA levels of proliferation, epithelial–mesenchymal transition, and angiogenesis markers in MDA-MB231 cell line. The data represent as means ± SD for the three independent experiments (* *p* < 0.05, compared with the control). (**B**) Effect of rutin (200 µM for 48 h) on the protein expression of above-mentioned markers. Blots shown correspond to observed effects in three independent experiments.

**Figure 6 nutrients-15-02884-f006:**
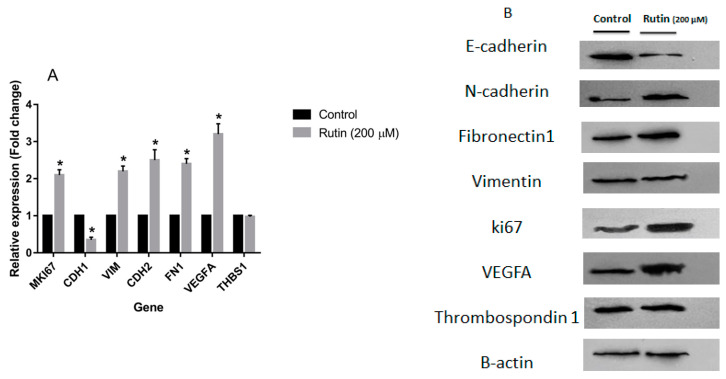
(**A**) The effect of ruin treatment at the concentration of 200 µM for 48 h on the expression of proliferation, epithelial–mesenchymal transition, and angiogenesis gene markers in MCF-7 cell line. The data represent as means ± SD for the three separate experiments (* *p* < 0.05, compared with the control). (**B**) Effect of rutin treatment (200 µM for 48 h) on the protein products of the mentioned markers in MCF-7 cell line. Blots shown correspond to observed effects in three independent experiments.

**Table 1 nutrients-15-02884-t001:** Primer sequences used in the present study.

Gene Name	Forward Primer	Reverse Primer
MKI67	5′-GCTACTCCAAAGAAGCCTGTG-3′	5′-AAGTTGTTGAGCACTCTGTAGG-3′
CDH1	5′-GGGGTCTGTCATGGAAGGTG-3′	5′-CGACGTTAGCCTCGTTCTCA-3′
CDH2	5′-GCGTCTGTAGAGGCTTCTGG-3′	5′-GCCACTTGCCACTTTTCCTG-3′
FN1	5′-ACAAGCATGTCTCTCTGCCAA-3′	5′-TCAGGAAACTCCCAGGGTGA-3′
VIM	5′-TCCGCACATTCGAGCAAAGA-3′	5′-ATTCAAGTCTCAGCGGGCTC-3′
VEGFA	5′-GAGCAAGACAAGAAAATCCC-3′	5′-CCTCGGCTTGTCACATCTG-3′
THBS1	5′-CCCTTGTGCTCAGAGTGGAT-3′	5′-GCCAGTAGAGAACAAATAAGCATGG-3′
GAPDH	5′-ACCCACTCCTCCACCTTTGA-3′	5′-CT GTTGCTGTAGCCAAATTCGT-3′

## Data Availability

Data are a available upon request to corresponding authors.

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
