# Peer review of "Rutin Promotes Proliferation and Orchestrates Epithelial–Mesenchymal Transition and Angiogenesis in MCF-7 and MDA-MB-231 Breast Cancer Cells"

_nutrients, 2023, doi:10.3390/nu15132884_

Round 1
Reviewer 1 Report
The article by Homa Hajimehdipour et al. entitled "Rutin promotes proliferation and orchestrates epithelial-mesenchymal transition and angiogenesis in human breast cancer cells." is quite interesting. However, it still raises the following issue.
1. All experiments were carried out using MCF-7 (ER-PR positive breast cancer cell lines) and MDA-MB-231 (TNBC cell lines). To validate the key findings of this study, additional ER-PR positive and TNBC cell lines (T47D, EFM19, E0771, 4T1, etc.) should be used.
2. Authors did not perform proliferation markers (Ki67, PCNA, etc), apoptosis and cell cycle markers and downstream and upstream pathways to validate rutin has potential as breast cancer promoting agents.
3. An additional control cell line (MCF-10) should be used to interpret and validate the key findings of the study
4. Authors did not confirm transwell and matrigel coated transwell assays for invasion and migration
5. Authors did not maintain consistency throughout the study in terms of rutin dosage. For instance, authors analysed EMT markers by western blotting at a concentration of 200 μM. Authors redo the experiments using 400 μM of rutin and confirmed its EMT promoting process based on its dosage dependent.
6. The authors did not perform any in vivo experiments in the present study and did not get consistent outcomes in the proliferative effect of rutin on tumor bearing animals
Moderate editing of English language is required
Author Response
Dear Reviewer,
We sincerely thank the reviewers for constructive criticisms and valuable comments, which were of great help in revising the manuscript. Accordingly, the revised manuscript has been systematically improved with new information and additional interpretations. Our responses (AC) to the referee’s comments (RC) are given below.
Reviewer 1:
“The article by Homa Hajimehdipour et al. entitled "Rutin promotes proliferation and orchestrates epithelial-mesenchymal transition and angiogenesis in human breast cancer cells." is quite interesting. However, it still raises the following issue”.
(AC) We would like to thank the reviewer for this nice comment.
(RC) “All experiments were carried out using MCF-7 (ER-PR positive breast cancer cell lines) and MDA-MB-231 (TNBC cell lines). To validate the key findings of this study, additional ER-PR positive and TNBC cell lines (T47D, EFM19, E0771, 4T1, etc.) should be used”.
(AC) We cordially appreciate the reviewer’s perspective and agree that using other breast cancer cell lines can bring new and interesting data regarding the effect of rutin on breast cancer cells. However, we should explain that we used these two different cell lines with different receptor expression characteristics as models to investigate the promoting effects of rutin on breast cancer cells with two different characteristics. We believe that this initially could be indicative of our hypothesis about the effect of rutin on breast cancer cells. Nonetheless, we will commit to designing future works to affirm this data using other cell lines. Although not requested by Reviewer 1, in order to clearly states that our study was conducted on MCF-7 and MDA-MB-231 as models of ER-PR positive and TNBC cells, the title was changed to “Rutin promotes proliferation and orchestrates epithelial-mesenchymal transition and angiogenesis in MCF-7 and MDA-MB-231 breast cancer cells”.
(RC) “Authors did not perform proliferation markers (Ki67, PCNA, etc), apoptosis and cell cycle markers and downstream and upstream pathways to validate rutin has potential as breast cancer promoting agents”.
(AC) We appreciate the reviewer’s perspective. We believe that we did not clearly explain this point in the original manuscript and this may have led to the present comments of the reviewer. Therefore, we should explain that we measured the mRNA and protein expression of a key proliferation marker Ki67 in both cell lines. The rationale for choosing ki67 was its role as a proliferation marker and its established role as a prognostic, metastasis, and predictive marker in patients with breast cancer. So, we selected it as a key marker in this study. Regarding the evaluation of other mechanisms or pathways, we should explain that the initial purpose of this present study was to evaluate the anti-metastatic effects of rutin on breast cancer cell lines because the literature shows that rutin is an anti-cancer agent, but our initial screening revealed opposite data. So, we went further to establish these observations by assessing its effects on proliferation, metastasis, and angiogenesis processes. Then, we decided to measure EMT and angiogenesis markers to further affirm obtained data.
(RC) “An additional control cell line (MCF-10) should be used to interpret and validate the key findings of the study”
(AC) With all due respect to the reviewer, we would like to explain that the MCF10A human mammary epithelial cell line is an in vitro model for studying normal breast cell function and transformation. However, the main goal of our study was to validate and prove the effect of rutin on breast cancer cell lines. Therefore, the purpose of the present study was not to study the effects of rutin on normal breast cells reason why MCF-10 cells were not used as control or in parallel.
(RC) “Authors did not confirm transwell and matrigel coated transwell assays for invasion and migration”
(AC) We would like to thank the reviewer for this suggestion. We agree that additional analyses would provide useful and important data, but we believe that the present analyses and findings are valid and important to reveal the effect of rutin on migration and invasion potential, at least on the two models cell lines used here. Indeed, the scratch assay used here has been validated in our lab and many other labs (and publications) as an authorized and robust way to investigate and identify molecules with effects on invasion and migration. In addition, the effect of rutin on the expression of markers of proliferation, EMT, and angiogenesis further confirm this observation.
(RC) “Authors did not maintain consistency throughout the study in terms of rutin dosage. For instance, authors analyzed EMT markers by western blotting at a concentration of 200 μM. Authors redo the experiments using 400 μM of rutin and confirmed its EMT promoting process based on its dosage dependent.”
(AC) With respect to the reviewer, we should explain that only proliferation and invasion assays were done using 200 and 400 μM of rutin. The rationale for this has lay in the MTT assay results. In the MTT assay, we saw that increasing concentrations of rutin increased the proliferation of both breast cancer cell lines. Therefore, we further decided to assess two upper limit concentrations of rutin for scratch assay. When we observed that both concentrations are effective in this assay on both cancer cell lines, we decided to do other molecular experiments by using rutin 200 μM, which was the lowest concentration of rutin with a significant effect on our previous functional experiments.
(RC) “The authors did not perform any in vivo experiments in the present study and did not get consistent outcomes in the proliferative effect of rutin on tumor bearing animals”
(AC) We would like to thank the reviewer for this suggestion. We agree that it will be important indeed to confirm our finding with an in vivo study. Unfortunately, we do not have such model in our labs. And creating a collaborative network to do it will be very long as animal testing requires months of administrative procedure before being possibly conducted.
(RC) “Moderate editing of English language is required”
(AC) We would like to thank the reviewer for pointing out the grammatical errors. We have revised the manuscript grammatically.
Regards
Dr M Maresca
Reviewer 2 Report
The study reported Rutin induced invasion and migration of cells in vitro. It was concluded that the mechanistic actions of Rutin involved the EMT process. More supportive data should be obtained to demonstrate the mechanistic actions. Also, why MDA-MB-231 and 16 MCF-7 breast cancer cell lines were used for the present study? The concluson can be elaborated based on more data that could demonstrate rutin may induce cancer cell cycle. Finally, more updated references on the subject matters should be cited.
A minor modification of English is recommended.
Author Response
Dear Reviewer,
We sincerely thank the reviewers for constructive criticisms and valuable comments, which were of great help in revising the manuscript. Accordingly, the revised manuscript has been systematically improved with new information and additional interpretations. Our responses (AC) to the referee’s comments (RC) are given below.
Reviewer 2:
“The study reported Rutin induced invasion and migration of cells in vitro. It was concluded that the mechanistic actions of Rutin involved the EMT process.
(RC) More supportive data should be obtained to demonstrate the mechanistic actions”.
(AC) We would like to thank the reviewer for this suggestion. Our data already demonstrate that rutin can stimulate proliferation, invasion and migration of breast cancer cells. And the mechanistic approaches conducted already demonstrate that rutin acts by modulating the expression of various genes/proteins involved in such processes. We agree that additional analyses would provide useful and important data about the signal pathway(s) involved in such effects. But we believe that the presented data already provided informations on how rutin can cause effects on breast cancer cells. Finding the precise signal pathways involved will be long and not sure of success. Therefore, we respectfully suggest to incorporate such long experiments into our future works on rutin.
(RC) “Also, why MDA-MB-231 and 16 MCF-7 breast cancer cell lines were used for the present study?”
(AC) We appreciate the reviewer’s perspective. The reason for choosing these two cell lines has lay in the different characteristics of these cell lines that are classically used as models in studies on breast cancer. MCF-7 cell line expresses estrogen and progesterone receptors but the MDA-MB-231 cell line doesn’t express any receptor and is known as a triple-negative breast cancer cell line. So, we wanted to evaluate the effect of rutin on receptor-positive and –negative breast cancer cell lines.
(RC) “The conclusion can be elaborated based on more data that could demonstrate rutin may induce cancer cell cycle”.
(AC) We would like to thank the reviewer for this suggestion. We agree that additional analyses would provide useful and important data, but we believe that the presented data about genes and proteins alterations already give informations on how rutin is acting on breast cancer cells.
(RC) “Finally, more updated references on the subject matters should be cited”.
(AC) With all due respect to the reviewer, we should assert that the present study has tried to cite more recent references and it is obviously evident in the reference list. However, the use of some old references (only references 26-29) was inevitable because there were no recent data regarding the phytoestrogenic effects of rutin.
(RC) “A minor modification of English is recommended”.
(AC) We would like to thank the reviewer for pointing out the grammatical errors. We have revised the manuscript grammatically.
Regards.
Dr M Maresca
Round 2
Reviewer 1 Report
Accept in present form
Moderate editing of English language required
Reviewer 2 Report
The revised looks acceptable.
minor editing is needed